# Advanced Optical Imaging-Guided Nanotheranostics towards Personalized Cancer Drug Delivery

**DOI:** 10.3390/nano12030399

**Published:** 2022-01-26

**Authors:** Madhura Murar, Lorenzo Albertazzi, Silvia Pujals

**Affiliations:** 1Institute of Bioengineering of Catalonia (IBEC), The Barcelona Institute of Science and Technology (BIST), 08028 Barcelona, Spain; mmurar@ibecbarcelona.eu (M.M.); l.albertazzi@tue.nl (L.A.); 2Department of Biomedical Engineering, Institute for Complex Molecular Systems (ICMS), Eindhoven University of Technology, 5612 AZ Eindhoven, The Netherlands

**Keywords:** nanomedicine, theranostics, optical imaging, personalized medicine, cancer

## Abstract

Nanomedicine involves the use of nanotechnology for clinical applications and holds promise to improve treatments. Recent developments offer new hope for cancer detection, prevention and treatment; however, being a heterogenous disorder, cancer calls for a more targeted treatment approach. Personalized Medicine (PM) aims to revolutionize cancer therapy by matching the most effective treatment to individual patients. Nanotheranostics comprise a combination of therapy and diagnostic imaging incorporated in a nanosystem and are developed to fulfill the promise of PM by helping in the selection of treatments, the objective monitoring of response and the planning of follow-up therapy. Although well-established imaging techniques, such as Magnetic Resonance Imaging (MRI), Computed Tomography (CT), Positron Emission Tomography (PET) and Single-Photon Emission Computed Tomography (SPECT), are primarily used in the development of theranostics, Optical Imaging (OI) offers some advantages, such as high sensitivity, spatial and temporal resolution and less invasiveness. Additionally, it allows for multiplexing, using multi-color imaging and DNA barcoding, which further aids in the development of personalized treatments. Recent advances have also given rise to techniques permitting better penetration, opening new doors for OI-guided nanotheranostics. In this review, we describe in detail these recent advances that may be used to design and develop efficient and specific nanotheranostics for personalized cancer drug delivery.

## 1. Nanotheranostics: Premises and Prospects

### 1.1. Need for Theranostics

Theranostics, as the name suggests, is the combination of specific diagnosis and targeted therapy in a single formulation. It is an upcoming field of medicine in which drug molecules and imaging agents are combined to allow simultaneous or sequential diagnosis and treatment of a disease [1]. The therapeutic agents include one (or more) drug or chemotherapeutic molecules, siRNA, or other oligonucleotides, while the diagnostic agents comprise image-contrast molecules or fluorescent probes that facilitate better resolved visualization of the diseased site with specificity. This property of having diagnosis and therapy in one package is proving to be a game changer for medicine as we know it [2]. Not only does it save time and money, but it also helps to avoid some of the undesirable side effects of conventional systemic approaches. With a view to evolve from conventional toward a more PM approach, theranostics aims to make healthcare more patient oriented [3]. Cancer is a complex case, in that it is not simply one disease, but a heterogenous group of diseases, predominantly characterized by uncontrolled growth and proliferation. Owing to this dynamic nature, a single type of treatment is rendered ineffective for overall patient population [2,4]. Chemotherapy, in particular, has a limited utility, as it is known to provide a more systemic response instead of a specific one, consequently giving rise to many side effects. Further, it also leads to treatment resistance during therapy, resulting in disease relapse [5]. Thus, there is a clear need for developing a theranostic path toward cancer PM.

The delivery of drug molecules at the target site, with simultaneous monitoring of treatment responses, provides customized feedback. Consequently, strategies such as drug dosage to be administered, patient management protocols and follow-up regimens can be adjusted to better suit the changing needs of each patient [6]. In this way, theranostic approaches can contribute to substantially enhance PM. The possibility of integrating molecular imaging along with therapy helps to identify and select a certain population of patients with a specific molecular phenotype (or biomarker) [7]. This further sheds light on the stage of the disease and also indicates the nature of response (either positive or negative) to the type of treatment provided [8]. Additionally, this approach proves to be a valuable tool in selecting a safe and effective dose and identifying adverse effects at early stages of therapy, while monitoring therapeutic response in real time and planning follow-up therapies [9]. The advantages of employing such a strategy are vast, and it promises to make great strides in cancer management [3,7]. Over the last decade, the theranostics paradigm has been moving toward employment of nanomaterials, thereby giving rise to “nanotheranostics” [10]. Nanoparticles (NPs) possess significant advantages (such as small size, biocompatibility, as well as their multifunctional ability, among others) that make them suitable for employment in theranostic medicine [11].

### 1.2. Role of Nanomaterials in Nanotheranostics

The application of nanotechnology in medicine is known as nanomedicine, and it comprises an interplay of a wide range of medical and scientific strategies [5]. One such approach involves the use of NPs in theranostics. Recent advances in nanomaterials are fueling the development of many nanotheranostic agents with both therapeutic and imaging functions [12,13]. The use of nanoscale materials imparts numerous advantages for both diagnosis and treatment, further leading to the development of nanosensors and nanomedicines, respectively [14]. However, it is important to note that the integration of two modalities in a nanosystem calls for a multi-parametric approach, involving the optimization of various factors (such as concentration of dyes, drugs and targeting moieties) in order to achieve efficient diagnosis and therapy [12].

NPs exhibit certain fundamental features that make them useful in a wide range of applications. Owing to their small size range of 1–100 nm, NPs have the benefit of localizing toward the disease sites in vivo, especially in the case of cancer [15]. For instance, their nanometric size precludes them from an easy renal clearance, thus allowing a longer circulation time in the blood in vivo, as compared to conventional chemotherapeutics. This further increases the chances of NPs to extravasate from tumor blood vessels and into tumor tissues, on account of an altered anatomy of tumor vasculature system, a phenomenon known as enhanced permeation and retention (EPR) effect [1,16,17]. They also possess a high surface-area-to-volume ratio that provides them with an increased functionalization potential for imaging probes, targeting ligands and other therapeutic agents. By conjugating the surface of NPs with active targeting ligands (such as antibodies, peptides or aptamers), they are able to accumulate at tumor sites and become subsequently internalized by cancer cells, thus accounting for targeted drug delivery [15,18]. Once the site of a disease has been identified by imaging approaches, NPs may also provide information on the extent or stage of the disease and further indicate disease response to treatment [14]. Nanotheranostics facilitate selective delivery of the conjugated or entrapped drug molecules at the target site, while monitoring their effective release in real time. Thus, they prevent the non-specific accumulation of drugs in potentially endangered healthy tissues, striving to improve the balance between efficacy and toxicity of systemic chemotherapeutic treatments [11,19,20].

NPs are known to have widespread applications in cancer treatment and have also received clinical approval for some applications for over two decades now [21]. There exist some, albeit few, examples of nanomedicines that have been approved for clinical use. Doxil, the first nanostructured formulation for the treatment of Kaposi’s sarcoma, received FDA approval in 1995 and was later also employed in metastatic ovarian cancer and multiple myeloma treatments. It consisted of polyethylene glycol (PEG)-coated liposomes and doxorubicin (DOX) encapsulated inside [22]. Coating nanomaterials with an inert polymer, such as PEG, is a popular approach in drug delivery, since it prevents the interaction of nanomaterials with blood components, thereby imparting “stealth” properties [23]. Consequently, the layer of PEG was responsible for imparting long-circulation properties. Additionally, this offered enhanced tumor accumulation and reduced off-site effects, which further helped reduce the adverse effects of DOX [22]. Doxil approval was then followed by other cancer nanomedicines entering the market, such as Abraxane (albumin-bound paclitaxel NPs), Marqibo (vincristine sulphate liposomes), DaunoXome (daunorubicin citrate liposomes), other DOX-containing liposomes, including Caelyx and Myocet [24,25], and many more.

### 1.3. Nanotheranostics in Cancer Treatment

As discussed thus far, NPs are the preferred delivery vehicles for exploitation in theranostic applications. They can be directed selectively to tumors by the processes of passive or active targeting. As mentioned earlier, the process of passive targeting involves extravasation of the NPs from blood vessels because of their altered anatomy and consequent accumulation at the tumor site selectively via the EPR effect [15]. On the other hand, NPs can be decorated with ligands, with strong affinity to bind specifically to targeted tumor cells, thus contributing to active targeting [26,27]. A study comprising multivariate analysis of the effect of different parameters (such as material, size, shape, surface coating, charge and targeting strategy) on the drug delivery efficiency of NPs, showed that active targeting had a delivery efficiency of 0.9%, as compared to 0.6% for passive targeting [28]. This demonstrates that active targeting can be a better approach for targeted drug delivery using NPs. Furthermore, a single nanoparticle can be labeled with one or more copies of ligands, or multiple ligands with different targets, for an efficient cellular uptake and enhanced tumor targeting [8,15].

Several nanosized carriers have been studied for theranostic applications. For example, studies involving the use of multi-functional gold and iron oxide NPs have been carried out to achieve a combined anti-cancer therapy using PET and MRI, respectively [29,30]. Quantum dots (QDs) possess inherent fluorescence and have also been explored for image-guided therapies [31]. Another widely studied inorganic material, known as carbon nanotubes, has also been a potential candidate for synchronous OI and drug/gene delivery [32]. Furthermore, polymeric NPs, owing to their additional feature of biocompatibility and drastically reduced toxicity, are widely explored in various theranostic applications [33].

Despite the various compositions of functionalized NPs explored for cancer theranostics, the ultimate aim is to reach the target site and achieve better efficacy of diagnosis and therapy, with a view to achieving improved pharmacokinetics and biodistribution of delivered molecules, thus increasing their stability and circulation time in blood [34]. For these purposes, the concept of developing nanotheranostics fits right in. Although there are no approved nanotheranostics on the market yet, there are several cancer nanomedicines with theranostic applications under different stages of clinical trials. Table 1 highlights those that involve nanotheranostics in cancer at various stages of development. Overall, we see that different types of nanomaterials are being employed for theranostic applications in a variety of solid tumors in a clinical setting. The bench-to-bedside translation of these nanotheranostics, however, still has its own set of challenges that need to be addressed by optimizing the right diagnosis technique and therapeutic agent.

The diagnosis procedure of cancer takes into account numerous factors: the patient’s symptoms, medical history and a physical examination. Nonetheless, a confirmation or an analysis report can be obtained by performing laboratory tests, a biopsy and imaging procedures. Biomedical imaging facilitates not only detection but also identification of the stage of disease, in addition to prognostic measures for the patient [35]. In addition, during treatment, imaging procedures allow for real-time monitoring of the response to therapy, thereby helping to demonstrate the efficacy of the chosen strategy [36].

Currently, the techniques routinely employed for molecular imaging include MRI, PET, CT, SPECT and OI [37,38]. These techniques are either used individually or in combination to provide synergistic imaging platforms. An early diagnosis plays a crucial role in obtaining a positive outcome, and so, these imaging techniques extensively contribute to the success of a therapy [36]. Even though MRI, CT and PET are well-established techniques and are used chiefly for development of nanotheranostics, recently advanced OI strategies are now offering certain advantages and properties that make them better suited for nanotheranostics and PM applications.

## 2. Optical Imaging as a Tool for Nanotheranostics

As discussed above, there are a variety of imaging techniques available that facilitate the diagnostic arm of nanotheranostics. However, each modality has its own advantages and limitations that need to be considered while choosing the most suited technique to obtain desirable outcomes. Briefly, MRI offers high spatial resolution and soft tissue contrast without tissue-penetrating limitations and is widely used in hospitals, but it is expensive, time consuming and relatively less sensitive. Radionuclide (PET/SPECT) imaging is also commonly used in clinical settings and offers high sensitivity with unlimited tissue penetration. However, it is expensive and provides limited spatial resolution compared with MRI. CT offers high spatial resolution and deep tissue penetration but imposes labor hazards, such as exposure to ionizing radiations. Fluorescence OI facilitates high-throughput screening for target confirmation and compound optimization, along with high sensitivity and multi-color imaging, but it has low penetration depth and spatial resolution. The advantages and disadvantages of different imaging modalities have been summarized in Figure 1. For a detailed comparison of various imaging techniques, see [39].

OI offers a non-invasive way of looking inside the body by using visible light to obtain images of organs and tissues. This is one of the biggest advantages of OI that makes it rather easy to use, as compared with other conventional techniques. Additionally, it is a relatively less expensive technique. In OI, intrinsic tissue absorption and scattering gives information about the anatomical characteristics but is not as informative about the specific functionality (e.g., metabolism, excretion and secretion) unless fluorescent markers are used. Further, due to its limitations, such as lower penetration depth and autofluorescence, OI has been used mainly in scientific research and less in clinical studies [40]. However, recent advances have improved penetration capability of OI, thereby improving its future clinical utility. Here, we will discuss the potential of fluorescence imaging as a promising tool for nanotheranostics. 

As already mentioned, fluorescence imaging must deal with two strong limitations: autofluorescence and low penetration depth. The former, though, can be exploited as a benefit in imaging. There have been studies where autofluorescence signals were used in endoscopic detection of esophageal neoplasia [41], and also similar devices were used for controlling neoplastic changes in oral, cervical and pulmonary mucosa [42,43,44,45], where neoplastic lesions were associated with a loss of autofluorescence. Regarding the limited penetration depth, many internal organs remain elusive, but many tissues/organs can still be reached [46,47]. There are studies in which examples have been provided for imaging the oral cavity, superficial lesions, breast (mammography-like imaging) [48,49,50,51,52,53], applications in ophthalmology cervix, trachea and gastrointestinal tract [54,55,56], skin, prostate, brain and lymph nodes [57,58]. Moreover, live imaging facilitates higher throughput as compared to classical dissection, thus providing faster analysis. In addition, it allows continuous imaging of the same animal at different time intervals in comparison to histology analysis where an animal is sacrificed at each time point.

Conventional fluorophores limit fluorescence live imaging to the surface (up to 1–2 mm), where the techniques of choice include confocal laser scanning imaging, multiphoton imaging, microscopic imaging by intravital microscopy or total internal reflection fluorescence microscopy. Near-infrared probes allow for a deeper imaging (up to several cms) and are the most promising choice for in vivo imaging [59]. Two-dimensional fluorescence reflectance imaging is mostly used, but it can only be used for superficial tissues and subcutaneously inoculated tumors, while fluorescence molecular tomography (FMT) [60], a 3D OI technique for NIRF-labeled probes, is capable of quantification and has a promising imaging depth maximum of around 12 cm [61]. However, one major setback it faces is the inability to accurately assign the organ from which the signal is obtained. Thus, micro-computed tomography is often coupled in order to obtain information about the site from where the signal comes [62].

From the recently developed super resolution microscopy techniques, stimulated emission depletion (STED) is the most promising one for in vivo imaging owing to its 3D sectioning capability. In that regard, live mouse STED imaging of cortical structures has been achieved recently [63,64]. In addition to tissue/organ imaging for diagnostics, fluorescence based OI is also favorable for surgical guidance. Improved tumor resection directly impacts on patient survival rates by decreasing tumor recurrence. This is particularly observed in the case of the gliomas, where the extent of removal of tumoral tissue during surgery is extremely crucial, as removing more than is necessary could lead to fatal secondary effects and is therefore a major prognostic factor [65,66].

Another possibility for fluorescence imaging involves ex vivo tissue samples from biopsies. Here, there is no penetration depth limit and so even the more sophisticated super resolution microscopy techniques can be applied [67,68,69,70]. Considering the three main applications discussed above (in vivo imaging, tissue section imaging and surgical guidance), OI can benefit from nanomaterials as a versatile carrier system integrating different moieties and allowing for an integrative nanotheranostics platform for patient diagnosis, staging and treatment follow up. 

### 2.1. Nanomaterials as Probes for In Vivo Optical Imaging

Nanomaterials are the ideal choice for integrating non-invasive imaging and treatment. Nano-sized materials embrace improved circulation, targeting ability, higher drug loading capacity and controlled drug release capabilities. Moreover, multiple functionalities can be easily combined in one platform, thus it is easy to fine tune and go toward PM [71]. Even though nanomaterials allow for simultaneous imaging and targeted drug delivery, certain tools are required for proper screening, as once they are injected in the body, they undergo many undesired transformations (such as protein corona formation and immune response) and need to cross many barriers before reaching the desired target. Screening the right formulation is usually carried out in vitro in cell monolayers, leading to an unexpected in vivo performance, which is typically worse than anticipated [28]. In this context, in vivo fluorescence imaging could be ideal for assessing nanomedicines formulation and biodistribution and consequently allowing for selection of the right formulation. 

Nanomaterials incorporate the imaging functionality, usually by covalent attachment of conventional fluorescent dyes, unless they are intrinsically fluorescent nanoparticles. Not only quantum dots, but other types of intrinsically fluorescent nanomaterials have been recently described. Carbon dots are physicochemically and photochemically stable nanoparticles, with strong fluorescence, low cost and low toxicity [72,73,74]. Black-phosphorus quantum dots [75,76,77] and lanthanide-doped upconverting nanoparticles (UCNps) [78,79,80] have also been proposed, with excellent optical properties. The targeting moiety is conjugated on the surface of the NP, and drugs can be encapsulated or covalently attached depending on the formulation of choice. Parameters such as stability, specificity, contrast and toxicity, must be taken into consideration when assessing the best formulation. It is noteworthy that fluorescent probes have much more stability as compared to radioisotopes and, therefore, allow imaging over longer time intervals. Additionally, fluorescence imaging can not only provide information on morphology but also functional and molecular properties; fluorescent probes for metabolic activity and smart fluorescent probes have been described. Smart fluorescent probes not only accumulate in the target tissue but are fluorescent only in the presence of the biomarker [81,82]. Interestingly, these smart probes can not only rely on enzyme activity but also gene expression, promoter activity, cancer cell tracking or vascular and lymphatic vasculature imaging, thereby expanding the imaging functionality toolbox [47]. This not only allows imaging at the tumor site but also gives information about disease progression that further directs the decision of best-suited treatment of choice.

Even though in vivo imaging represents a more reliable screening platform for nanomedicines, checking one formulation at a time limits the number of formulations to be tested. This is where the multicolor capability of fluorescence imaging comes into play, providing a better alternative. Barcoding is a process that allows for simultaneous screening of a library of NPs. By using different fluorophore combinations, different emission spectra are obtained [83,84], which are later analyzed by linear unmixing to differentiate each population. Then, a mixture of NPs with different formulations and/or properties can be tested to see which ones reach the target site, which is elucidated by a specific color code. Similar work, but in an in vivo setup and with DNA barcoding, has been recently established, as displayed in Figure 2 (below) [85]. Further, regarding multiplexing in an in vivo setup, it has already been demonstrated how three different fluorophores can be detected in a mouse colon by endoscopy [86,87]. This multiplexing ability, once again, highlights the potential of nanomedicine combined with advanced OI in going toward PM; the choice of best-suited formulation for individual patient, obtained in a single test via high throughput screening.

So far, FMT has been used to visualize different types of nanomaterials. Examples include NIRF-labeled high-density lipoprotein NPs to image active vs. passive tumor targeting over time [88], cyanine-containing chitosan-based nanocarrier accumulation on SCC7 xenografts (squamous cell carcinoma) [89], mesoporous silica NPs distribution in mice bearing metastatic 4T1 tumors (tumor sentinel lymph node) [90] or NIRF-labeled polymeric drug carrier passively targeting CT26 colon carcinoma xenografts [62]. RGD peptide-labeled QDs have also been used for cancer targeting and imaging, especially for imaging α_V_β_3_ integrin positive tumor vasculature [91]. Many preclinical studies have been described with QDs, for instance targeting epidermal growth factor receptor (EGFR) with multifunctional siRNA-QD constructs for selectively inhibiting the expression of EGFRvariant III in target human U87 glioblastoma cells [92].

### 2.2. Imaging-Guided Surgery and Drug Delivery

Tumor extraction surgeries could benefit from fluorescence imaging, by use of selective labeling of the tumor cells that acts like a guide for the surgeon. The role of fluorescence imaging in image-guided surgery has been proven useful when QDs were injected in mice to monitor lymphatic drainage with potential use in cancerous nodes resection [93]. Activatable cell-penetrating peptides have also been described for surgery-guided resection [94,95,96]. In glioblastoma, QDs can also be used as intraoperative assistance for distinguishing healthy and unhealthy tissue [31,91,92,97,98]. In imaging tissue sections, QDs decorated with five different biomarkers have been used for imaging breast tumor sections [99].

Another important aspect is that nanomaterials allow for multimodal imaging, and many examples of combining OI with one of the following techniques have been described: MRI, photoacoustic imaging, spectrally enhanced Raman Spectroscopy imaging, PET or SPECT. As explained, CT is usually coupled with FMT to provide the reference space lacking in FMT. One example is the biodistribution and tumor accumulation of polymeric nanocarriers [62]. The information obtained from multimodal imaging of nanotheranostics will allow fine tuning of the drug therapeutic dose, while simultaneously monitoring the progression of the tissue of interest, treatment efficacy and kinetics delivery. This will not only lead to an early diagnosis but also help eliminate under/over dosage of drugs [38,100]. In addition, image-guided drug delivery can be coupled with drug discovery for identifying biomarkers of drug efficacy and safety while understanding disease processes, thus reducing the development time [61]. An example would be a polymeric nanotheranostic system for both OI of breast cancer progression and drug release, as drug and fluorophore are released by enzymatic cleavage performed at the same time [101].

As mentioned already, in vivo fluorescence microscopy will elucidate the right formulation for each case study, thus the focus can be kept on the treatment. Therefore, once the nanosystem platform has been established, fluorescence imaging can be used as a tool for non-invasive visualization of drug distribution, release kinetics, accumulation, drug therapeutic effect and elimination [62]. A few examples of the use of in vivo fluorescence imaging for medical imaging discussed in the text above have been demonstrated in Figure 3 below. Indeed, it is with translational imaging that we can move toward individualizing nanomedicines [60].

## 3. Nanotheranostics for Personalized Medicine

### 3.1. Need for Personalization

As explained before, the paradigm of cancer medicine is changing. It is slowly progressing toward use of highly precise theranostics, even at the molecular level. Consequently, the field of personalized or precision medicine has evolved as a new trend in cancer medicine, holding promise of improving healthcare before, during and after the disease. It has been developed due to the understanding that the “one-drug-fits-all” approach, wherein a single therapeutic agent shows similar outcomes in many patients with the same type of disease, no longer stands true, especially in the case of a dynamic disease, such as cancer [14,71]. Instead of the conventional practice of most common prescription, the new trend is to develop precise theranostic procedures that are designed, considered and tailored to individuals at the genetic level for the most appropriate treatment [102].

Moreover, with the help of genome sequencing, PM can further aid in determining the patient’s susceptibility to a disease, thus enabling monitoring and disease-prevention regimens [14]. Thus, for an efficient development of PM, the diagnostic testing of key molecules involved in a disease is extremely pivotal. The advent of PM also promises to redirect the success of pharmaceutical drug development and research toward patients who are genetically identified as positive “responders” to the compound [71]. Even though there is no defined market for PM in the private sector yet, the National Institute of Health (NIH) and the US FDA have developed several NIH-supported centers and public–private partnerships to translate potential candidates toward the clinic [6].

Nanotheranostics favor the collective efforts of diagnostic imaging and therapy into one nanoscale system, and therefore, fit directly into the concept of PM [103]. The amalgamation of therapeutic drugs and imaging agents in a nanoplatform allows for the screening of patients and assigning them to a set treatment regime and studying the progression of tumor post treatment with the help of contrast agents [104]. The selection of patients can be carried out in a multistep process, using a combination of genome sequencing, diagnostics and image-guided therapy [105]. In conclusion, a model PM nanotheranostic system for cancer should first be able to identify the cancer type, image the diversity of tumor, employ a tailor-made treatment based on the diagnostic and imaging results and, lastly, examine treatment efficacy.

### 3.2. Progress So Far: Nanotheranostics in Clinical Trials

In the age of PM, modern nanomedicine is focused more on tumor heterogeneity and is directing toward tailoring of treatment regimens for individual patients. The utility of nanotheranostics in this movement is highly application oriented, so it is vital to exploit its maximum potential in order to aid the development of more efficient cancer treatments [106]. The rapid advancement in the field of diagnosis or imaging can be seen clearly in the continuous increase in the number of imaging-based clinical trials listed by the US NIH, which started from only about 120 studies that began in the late 1990s, increasing to more than 6000 trials that were launched already in the last decade. However, very few of them involve nanotheranostics, whereas the majority are found to employ small-molecule-based molecular imaging for different applications. Even though nanotheranostics hold potential to revolutionize PM, they are currently far from full clinical utility [107]. Many different types of small molecules or ligands have been tested for theranostic applications. Some examples of cases of small molecules having similar characteristics, such as nanotheranostics that are under clinical trials, are highlighted in Table 2.

## 4. Discussion and Future Perspectives

The development of nanotheranostics for an effective cancer therapy involves a multi-disciplinary research and collaborative efforts from the fields of material sciences, cancer biology and imaging sciences. NPs can be suitably used as systems for the construction of a convenient “all-in-one” platform that encompasses all the necessary (diagnosis, targeting, controlled drug release and therapy) functions. The ultimate goal of theranostic nanomedicine research is to use nanotheranostic platforms to customize treatment for a disease and individual, patient-specific needs. Several nanotheranostics and nanomedicines are under clinical trials, as discussed in this review. However, there are still a few critical issues remaining that need to be addressed for a successful clinical translation of nanotheranostics. 

The choice of the most favorable combination of therapeutic and imaging modality is extremely crucial, as they both have their own strengths and weaknesses. Optimizing the level of therapeutic and imaging agents simultaneously in a single system might also pose a challenge. Furthermore, the design of a nanotheranostic formulation calls for thorough optimization, such that neither component (imaging/ therapeutic moiety) is released from the delivery system prematurely, thus providing correct information about its therapeutic outcomes. In addition, scaling up the synthesis of tested nanotheranostics is challenging, however necessary. For most nanotheranostic systems developed so far, their safety in humans has not been studied extensively.

Furthermore, we are now heading toward personalization of therapy as per patient’s requirements, which entails generation of tremendous amounts of information (the so-called “-omics” data) even at the molecular level. This is where artificial intelligence (AI) can be harnessed for processing this huge amount of data. For the purpose of personalization, the concept of combinatorial nanomedicine is also gaining more popularity. It involves the synergistic co-delivery of more than one drug that further promises markedly improved outcomes, patient accessibility to affordable nanomedicines and more drug approvals. Recently, AI was employed to optimize this and help pinpoint to globally optimized drug combinations. Additionally, it was used for dynamic monitoring of nanomedicines for drug delivery and subsequent modulation of treatment [120]. Therefore, the future of personalized nanomedicine does foresee the utilization of AI to overcome challenges, such as the optimization of nanomedicines, as well as drug development pipelines. It can also be employed for AI-guided therapy in the clinic.

To summarize, a deeper understanding of the interactions between nanotheranostic system and the human body, an elaborate and long-term assessment of its toxicity and an establishment of regulatory protocols is required for the development of more efficient nanotheranostics. As we gain more understanding of the role of advanced OI for screening these systems, new possibilities for developing more efficient and individualized nanomedicines become feasible, paving a way toward PM. As said before, we believe that advanced OI techniques can be exploited to screen prospective libraries of nanomaterials simultaneously and in real time.

## Figures and Tables

**Figure 1 nanomaterials-12-00399-f001:**
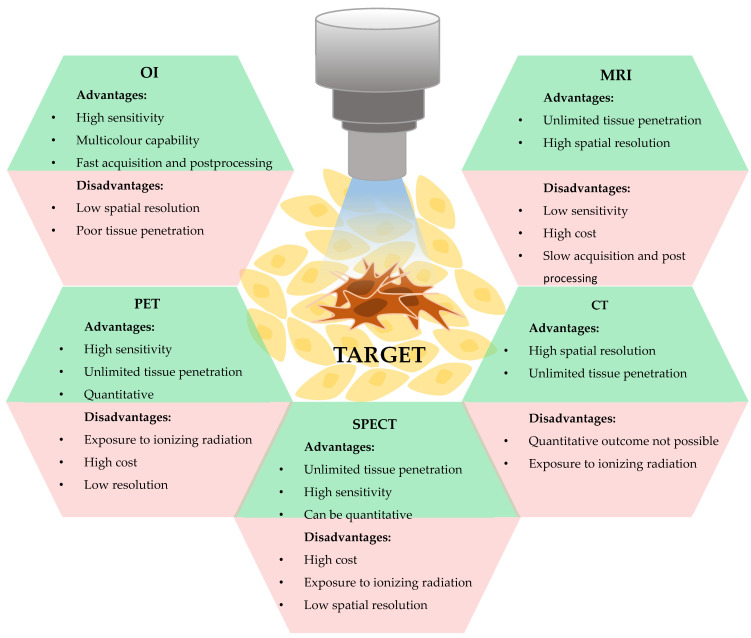
Advantages and limitations of different biomedical imaging techniques.

**Figure 2 nanomaterials-12-00399-f002:**
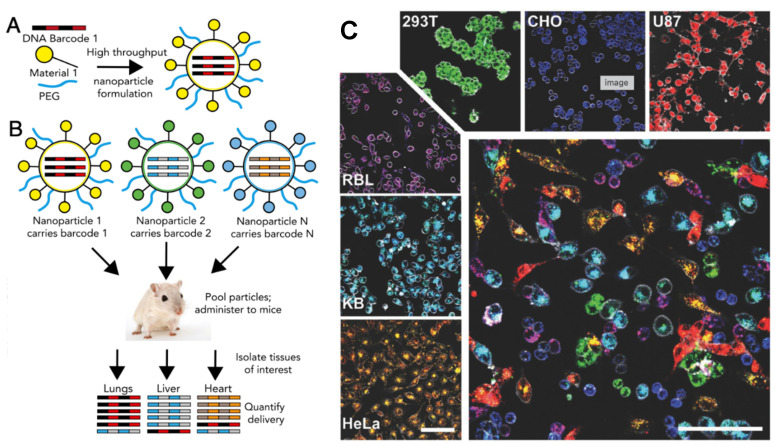
Multiplexing of different optical probes for simultaneous screening of library of NPs targeting cancer cells in vivo (**A**,**B**) and in vitro (**C**). Adapted with permission from [83,85] respectively; Copyright, 2017 John Wiley & Sons, Inc.; Copyright, 2017 National Academy of Sciences.

**Figure 3 nanomaterials-12-00399-f003:**
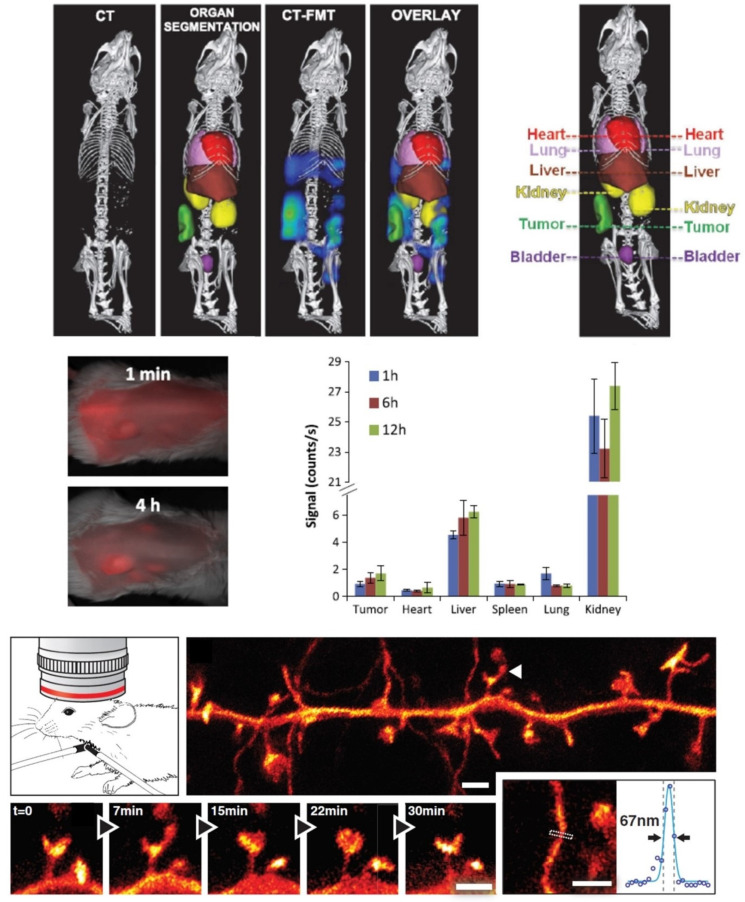
Examples of in vivo fluorescence OI for medical imaging. Adapted with permission from [62,63,101] respectively; Copyright, 2013 ACS Nano; 2012 Science; 2014 Cancer Lett.

**Table 1 nanomaterials-12-00399-t001:** Studies of clinical trials involving nanotheranostics in cancer.

Nanomedicine	Condition	Phase	Clinical Trial Status	Imaging Modality Employed	Clinical Trial Identifier *
Lyso-thermosensitive liposomal doxorubicin (LTLD, ThermoDox)	Stage IV breast cancer	1	Ongoing	Magnetic resonance guided-high intensity focused ultrasound (MR-HIFU)	NCT03749850
Vincristine liposome	CD20+ aggressive B-cell lymphoma	3	Recruiting	Fluorodeoxyglucose Positron emission Tomography (FDG-PET)	NCT01478542
Feraheme (SPION)	Pancreatic cancer	4	Completed	Ultra-small superparamagnetic iron oxide magnetic resonance imaging (USPIO-MRI)	NCT00920023
Silica NPs	Nodal metastases of neck melanoma, colorectal and breast cancer	1 and 2	Recruiting	Real-time OI using fluorescent cRGDY-PEG-Cy5 5-C dots	NCT02106598
89Zr-nanocolloidal	Colon cancer	2 and 3	Completed	PET/CT and intraoperative near infra-red fluorescence (NIRF) imaging	NCT02850783

* Information retrieved from clinicaltrials.gov, accessed on 1 July 2021.

**Table 2 nanomaterials-12-00399-t002:** Studies relating to cancer nanotheranostics under clinical trials.

Implication	Nanotheranostics (Small Molecule) Employed	Specification	Application	References
Nanotheranostics with “switchable” properties that can selectively mark specific tissues, such as tumors or inflammation	5-Aminolaevulinic acid (5-ALA)	5-ALA, an endogenous precursor of haemoglobin that produces porphyrins (which fluoresce under violet blue light illumination) in some types of malignant brain tissues	Intraoperative fluorescence-guided complete resection of several brain tumors	[108,109,110,111]
Nanotheranostics for active targeting	Folate–fluorescein isothiocyanate conjugate. Super paramagnetic iron oxide (SPIO) NPs	Folate receptor α (FR-α) is overexpressed in 90–95% of epithelial ovarian cancers and is therefore a good candidate for active targeting These particles possess excellent biocompatibility and magnetic properties and have been widely used for drug delivery, MRI probes, and tumor thermotherapy	A FR-α-targeted fluorescent agent used for intraoperative fluorescence imaging guided ovarian cancer surgery. SPIO-enhanced MRI mapping of liver cancer, non-invasive imaging of lymph node metastases, active targeting of lung cancer,	[112,113,114,115,116] (p. 2)
Ligand targeted nanoparticle drug conjugate (NDC)	Cornell (cRGDY-PEG-Cy5-C) dots conjugated to ^124^I radioisotope	^124^I-cRGDY-PEG-Cy5-C dots are sub-10nm fluorescent core shell silica NP targeted to αV/β3 integrin receptor overexpressed on angiogenic endothelial cells and on various cancer cells	dual-modality (PET–OI) imaging for targeted molecular imaging of integrin-expressing cancers	[117,118,119]

## Data Availability

Not applicable.

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
