# Peer review of "Advanced Optical Imaging-Guided Nanotheranostics towards Personalized Cancer Drug Delivery"

_nanomaterials, 2022, doi:10.3390/nano12030399_

Round 1

Reviewer 1 Report

Nanomedicine provides enormous potential for personalized medicine (PM). Nanotheranostics combines diagnostic imaging and therapy, which are developed to fulfil the promise of PM by helping in selection of treatments, objective monitoring of response and planning of follow-up therapy. In this review, the author focuses on the optical imaging (OI) with high sensitivity, spatial and temporal resolution, and less invasiveness for PM. The paper makes a comprehensive introduction to the OI techniques and the related nanotheranostics. Therefore, this review is of practical significance and should be considered for publication. Nevertheless, I do feel that some points should be addressed in revised form before it tis submitted.

  1. Nowadays, lots of nanomaterials have been developed as probes for optical imaging. The author should summarize the progress in detail such as black phosphorus-based nanoagent (Nano Lett. 2019, 19, 8, 5587) and carbon-based nanosystem (J. Am. Chem. Soc. 2009, 131, 11308), etc.
  2. The challenges should be included in this review, which will provide more understanding for OI and PM.
  3. The whole manuscript contains several undefined non-standard abbreviations, please go through the entire text and check that all of them are defined when they first appear as well as in all the self-standing sections. (Ref 14…, etc) And the authors should added the number of the latest references.

Author Response

Dear reviewer,

We really appreciate your time and all the valuable feedbacks  that have helped to improve the review. Here we provide a comprehensive overview of the changes made in the revised draft of the manuscript.

  • Feedbacks from Reviewer 1 and comments from authors (in blue):
  1. Nowadays, lots of nanomaterials have been developed as probes for optical imaging. The author should summarize the progress in detail such as black phosphorus-based nanoagent (Nano Lett. 2019, 19, 8, 5587) and carbon-based nanosystem (J. Am. Chem. Soc. 2009, 131, 11308), etc. We have added a paragraph on new types of probes for optical imaging. Both references suggested by the reviewer and others have been added and discussed in section 2.1. (Nanomaterials for optical imaging).
  2. The challenges should be included in this review, which will provide more understanding for OI and PM. We discuss advantages and disadvantages of optical imaging in section 2 (from line 176 to line 246). They are also included in Figure 1. We believe the limitations of OI can be seen as its challenges.
  3. The whole manuscript contains several undefined non-standard abbreviations, please go through the entire text and check that all of them are defined when they first appear as well as in all the self-standing sections. (Ref 14…, etc) And the authors should add the number of the latest references. The entire manuscript has been proof-read and all abbreviations are defined when they first appear. In addition, the authors have provided a list of acronyms at the end of the draft (post the list of references). Several latest references have also been added.

We hope that they are in accordance with the insights provided by the reviewer, and in case of further clarification, we would be happy to answer all the doubts.

Reviewer 2 Report

In this manuscript, the authors aim to introduce and summarize the development of nanotheranostics which comprise both therapy and diagnostic imaging to customize treatment for disease and individual patient-specific needs (Personalized medicine, PM). The authors have summarize the current progress in nanomaterial-based nanomedicine, imaging technology and imaging-guided surgery and drug delivery systems, respectively, and looked forward to the use of artificial intelligence (AI) to meet the challenges of nanomedicine optimization as well as drug development pipelines. This manuscript is recommended to be published after major revision.

  1. There are so many mistakes in the text, such as the format errors, grammatical errors, and misuses of punctuation (e.g., line 73, line 88, line 173, line 192, line 220, line 227, line 255, line 260, line 262, line 298, line 32, line 359, line 394, line 486). Please check the whole manuscript carefully.
  2. The sub-headings in the review should be logical and conclusive. For example, given the title "1. Nanotheranostics: Premises and prospects" and "1.1 Need for theranostics", it is recommended to change the subtitle "1.2 Role of nanomaterials" as "1.2 Role of nanomaterials in theranostics", which may make the line of review clearer. All the sub-headings are recommended to be revised.
  3. In this review, the latest references cited in this manuscript were published in 2019. It is recommended to add literatures published in the past two years. In addition, the literatures published before 2015 should be updated.
  4. There are only two figures in this manuscript. To make the content more intuitive, it is recommended to provide more figures.
  5. DNA nanotechnology has become a powerful tool in bioimaging and biomedicine, which thus should be introduced in the review with the citation of the most published works.

Author Response

We really appreciate the time and all the valuable feedbacks from reviewer 2 that have helped to improve the review. Here we provide a comprehensive overview of the changes made in the revised draft of the manuscript.

  • Feedbacks from Reviewer 2 and comments from authors (in blue):

  1. There are so many mistakes in the text, such as the format errors, grammatical errors, and misuses of punctuation (e.g., line 73, line 88, line 173, line 192, line 220, line 227, line 255, line 260, line 262, line 298, line 32, line 359, line 394, line 486). Please check the whole manuscript carefully. The entire manuscript has been proofread for format, grammatical errors and punctuation. All the mistakes listed by the reviewer have been fixed.

  1. The sub-headings in the review should be logical and conclusive. For example, given the title " Nanotheranostics: Premises and prospects" and "1.1 Need for theranostics", it is recommended to change the subtitle "1.2 Role of nanomaterials" as "1.2 Role of nanomaterials in theranostics", which may make the line of review clearer. All the sub-headings are recommended to be revised. All sub-headings in the manuscript have been revised as recommended.

  1. In this review, the latest references cited in this manuscript were published in 2019. It is recommended to add literatures published in the past two years. In addition, the literatures published before 2015 should be updated. Latest references and older ones have been added.

  1. There are only two figures in this manuscript. To make the content more intuitive, it is recommended to provide more figures. Figure 2 has been added to help in the understanding of the text.

  1. DNA nanotechnology has become a powerful tool in bioimaging and biomedicine, which thus should be introduced in the review with the citation of the most published works. We have introduced the use of DNA nanotechnology for bar-coding different types of theranostic nanomaterials (around line 288 and Figure 2). Despite DNA nanotechnology is being widely used in in vitro applications, we have not found more examples on DNA nanotechnology for in vivo optical imaging.

We hope that they are in accordance with the insights provided by the reviewer, and in case of further clarification, we would be happy to answer all the doubts.

Round 2

Reviewer 2 Report

This manuscript has been well revised according to the reviewer's comments, which can be accepted in its present form.